# OpenReview forum: "MedMT-Bench: Can LLMs Memorize and Understand Long Multi-Turn Conversations in Medical Scenarios?"
_ICLR.cc/2026/Conference — ICLR 2026 Conference Withdrawn Submission_

### Official Review · Reviewer_FuhP · 2025-10-27

**Soundness:** 2
**Presentation:** 2
**Contribution:** 2
**Rating:** 2
**Confidence:** 4

**Summary:**

This paper mostly focus on a 'multi-turn medical conversation' in LLM evaluation as contribution. The eval mostly focuses on instruction following on various medical scenarios that the authors generated. Things like prohibiting unsafe output, detect ambiguous queries, multiple queries handling etc. The generated data has 400 test cases about 22 turn average. And LLM as judge is used for atomic test points (ATPs). Some efficiency gains are discussed by using synthetic multi-agent generation.

**Strengths:**

- The medical domain doesn't see much multi-turn conversation evaluation which makes the question at hand timely
- The benchmark coverage is adequate, where they used pretty much all the frontier and open-source models so the numbers provided can help community to differentiate some of the models

**Weaknesses:**

- Overall the benchmark is only focusing synthetic data, i.e. no real patient data, no real world driven scenario. Although there are some expert reviews, the contribution is incremental. Take Healthbench as example, all of the scenarios are real world driven and are sourced globally to uncover some of the common issues LLMs have in medical domain.
- Further, on some concepts, the authors equate “instruction following” with “medical safety and reasoning.” In high-stakes domains, those are distinct; a model can obey instructions yet make unsafe clinical inferences. This conflation limits theoretical clarity. The instruction following is solely deemed as the primary proxy for a model’s safety, reasoning, and reliability in medical contexts. This is a very narrow scope of the whole domain. For example, if the system prompt says “Do not provide drug brand names” and the model obeys, it is labeled “safe” — but it could still recommend a medically incorrect dosage or miss an urgent symptom, which the benchmark does not capture.
- ATPs agreement was measured on a small subset of instances (unclear N). Statistical uncertainty is not reported. For a technical conference these things should be a given
- Overall the benchmark may create a false sense of “medical readiness” since all dialogues are LLM-synthetic; true human–model interactive variance is not measured.

**Questions:**

- Why use Gemini-2.5-Pro as both evaluator and one of the evaluated models? It risks model-family bias
- What's the total number of rubrics across multi turn conversations?
- Can you give more details on "medical team"? The paper lacks quantitative evidence of inter-annotator agreement or number of clinicians involved. “Professional medical data team” is vague and doesn't lend any credibility if no evidence

---

> ### Author Response · Authors · 2025-11-21
> **Thank you so much for your comments! We have answered all your questions point by point.**
>
> Thank you so much for your comments! Thanks to your helpful suggestions, we have supplemented and improved our paper. Below are our responses to the questions you raised.
>
> > Q1. Overall the benchmark is only focusing synthetic data, i.e. no real patient data, no real world driven scenario.
>
> We have deployed large language models (LLMs) in real-world medical scenarios at scale, serving a substantial user base. **All evaluation dimensions in the benchmark were derived from issues observed in production use**.
>
> In view of privacy and security issues, we synthesize multi-turn dialogues that rigorously simulate real scenarios. **To ensure clinical plausibility and appropriateness, a professional medical team reviews the synthesized data for scenario validity and removes any noncompliant items.**
>
> Finally, it's worth mentioning that we examined instruction following capability. **Even with synthetic data that deviates somewhat from real-world scenarios, we were able to effectively assess the existing model's shortcomings in instruction following.**
>
> > Q2. Further, on some concepts, the authors equate “instruction following” with “medical safety and reasoning.”.
>
> We also believe it’s incorrect to equate instruction following with medical safety and reasoning. We offer the following clarifications:
>
> **Scope and intent:** The "medical safety and reasoning" we mentioned focuses on the area of ​​**instruction following**. In this benchmark, the objective is to detect such violations and respond appropriately (e.g., refuse or redirect with safe guidance), rather than to assess general clinical reasoning.
>
> Errors involving dosages, prescriptions, and other strictly clinical content are outside the scope of this instruction-following benchmark. We are concurrently developing this factuality-focused component and plan to release it in the near future.
>
> > Q3. ATPs agreement was measured on a small subset of instances (unclear N). Statistical uncertainty is not reported. For a technical conference these things should be a given
>
> We apologize for not clearly stating the experimental details for Table 2 in the original manuscript. **The consistency check was computed over the entire benchmark evaluated across 7 frontier models, totaling 2,800 samples**, rather than on 400 samples per single model. In addition, we have supplemented Table 2 with the mean and standard deviation across repeated evaluations. Under atomic test points, this fluctuation is bettween ±0.6 to ±1.2, indicating limited variability and stable outcomes with TP-guided evaluation.
>
> | Consistent rate   | Auto-Eval without ATPs| Auto-Eval with ATPs|
> |-------------------|-----------------------|--------------------|
> | GPT-5             | 40.33%                | 85.40% ±1.2        |
> | Claude-4-Opus     | 39.68%                | 86.97% ±0.8        |
> | OpenAI o3         | 37.98%                | 87.03% ±1.1        |
> | Gemini-2.5-Pro    | 41.09%                | 91.94% ±0.6        |
> | GPT-4o            | 39.91%                | 86.99% ±1.0        |
> | GPT-4.1           | 40.93%                | 88.96% ±0.8        |
>
>
> > Q4. Why use Gemini-2.5-Pro as both evaluator and one of the evaluated models? It risks model-family bias
>
> We presented a detailed comparison of the agreement rate between Gemini-2.5-Pro ​​and human assessments when evaluating responses from different models. The results show that, Genmini-2.5-Pro ​​did not exhibit self-bias in the correctness of its own responses.
>
> |       Models      | Consistent rate |
> |-------------------|-----------------|
> | GPT-5             | 89.50%          |
> | Claude-4-Opus     | 91.25%          |
> | OpenAI o3         | 92.25%          |
> | Gemini-2.5-Pro    | 93.00%          |
> | GPT-4o            | 94.50%          |
> | GPT-4.1           | 91.25%          |
>
> > Q5. What's the total number of rubrics across multi turn conversations?
>
> We have expanded **Section 3.3** to describe in greater detail the generation and validation pipeline for test points, and we have added two more detailed illustrative examples. Concretely, given a synthesized, task-specific multi-turn context, we generate in the final turn an instruction-following question targeting a specific evaluation dimension and, concurrently, its corresponding test point. Each multi-turn sample is paired with a corresponding test point.
>
> > Q6. Can you give more details on "medical team"?
>
> Specifically, we maintain a professional medical annotation team of approximately 20 full-time staff, all with master’s or doctoral degrees from medical institutions. In addition, this core team coordinates and oversees several dozen part-time annotators with medical knowledge, who perform manual evaluations.

---

> ### Author Response · Authors · 2025-11-27
> **Kindly Requesting Your Thoughts on Our Response**
>
> Dear Reviewer,
>
> Thank you very much for your time and valuable feedback on our submission. As we near the end of the rebuttal period, we would appreciate your thoughts on whether our response has sufficiently addressed your primary concerns. If there are any additional suggestions or clarifications, we are more than willing to engage further to improve the paper.

---

### Official Review · Reviewer_B8DY · 2025-10-30

**Soundness:** 2
**Presentation:** 2
**Contribution:** 2
**Rating:** 4
**Confidence:** 3

**Summary:**

The authors propose a new benchmark, MedMT-Bench, which has 400 samples, each with on average 22 turns. The benchmark asks questions along 5 key axis, including long context, self correction, and instruction clarification, with question coming from many different medical departments and scenarios. The authors propose adding atomic tests within each chat that are easier for the LLM auto-eval to judge, allowing the authors to get 92.18 percent agreement between human experts and Gemini 2.5 Pro.

**Strengths:**

I think the idea of the atomic test points is interesting, as a way to better use LLMs for automatic evaluation.

I also think it's a valuable contribution to create more open source evaluation data along the key dimensions in the paper.

The paper is testing a large range of models (17) which is good to understand which models perform well in this domain.

**Weaknesses:**

The most major weakness from my point of view of this work is that it's not possible to properly evaluate the contributions at in the paper as many details are missing, despite having a very long (60 pages) paper. In particular:

It's not clear if the accuracy metric is the fraction of the total test points that pass over the full corpus or if it's aggregated over the conversations.

Why does the models have a consistency of less than 50 % (random guessing if I understand the metric) without the atomic test points?

The atomic test points are insufficiently described in the main paper. How many are there per conversation? It's also unclear how many atomic test points the accuracy metric for each capability dimension represent.

There are no details on the human evaluation. Is this done on the full benchmark 400x(22 average turns). Who were the evaluators etc.

It would have been nice if there was an analysis how relevant/aligned performance on the test points was to performance on the downstream capabilities outlined in the paper.

Other weaknesses:

I think it's overselling the human-ai agreement when you report 92.18% as the human-AI agreement, as this is the highest agreement (human and Gemini 2.5 Pro) not the average between LLM and Human.

Please report the consistency rate for all the 7 models for which the human evaluation was conducted.

Minor: some relevant recent work on Multi turn conversations is not cited: MMMT-IF: A Challenging Multimodal Multi-Turn Instruction Following Benchmark

**Questions:**

Clarification: Does the benchmark task that the LLM is asked to do corresponding to a real LLM use case in health care? Are LLMs already applied in this setting?

Do we need a benchmark that combines multi-turn with complex instructions, at the same time as a medical domain specific one. Does it add value on top of having two evaluation benchmarks that already exist?

Please clarify how the consistency rate is computed, a simple equation would make it more clear.

Have you considered the bias that may come from evaluating Gemini 2.5 Pro with Gemini 2.5 Pro also as the judge (this has for instance been documented in the paper MMMT-IF: A Challenging Multimodal Multi-Turn Instruction Following Benchmark).

---

> ### Author Response · Authors · 2025-11-21
> **Thank you so much for your comments! We have answered all your questions point by point.**
>
> Thank you so much for your comments! Thanks to your helpful suggestions, we have supplemented and improved our paper. Below are our responses to the questions you raised.
>
> > Q1. It's not clear in the accuracy metric.
>
> We omitted this part of the explanation in the main text. We have added the following equation into **Section 5**.
>
> $$
> \text{Score} = \frac{1}{N} \sum_{i=1}^{N} \mathbf{1}(a_i \models c_i).
> $$
>
> where $N$ is the total number of the benchmark instances, $a_i$ is the model’s output, and $c_i$ is the associated evaluation criterion.
>
> > Q2. Why does the models have a consistency of less than 50% without the atomic test points?
>
> The automatic evaluator tends to lean toward answering “correct” without the atomic test points, and the distribution of **correct versus incorrect responses is imbalanced (approximately 45:55)**. So this class imbalance yields an overall agreement/accuracy below 50%.
>
> > Q3. The atomic test points are insufficiently described in the main paper.
>
> We have expanded **Section 3.3** to describe in detail and have added two more comprehensive illustrative examples.
>
> > Q4. There are no details on the human evaluation. Is this done on the full benchmark 400x(22 average turns). Who were the evaluators etc.
>
> Specifically, **the overall evaluation was conducted by 20 full-time professional medical masters/doctors, along with dozens of part-time medical professionals.** Each item is independently assessed by at least two annotators, and any disagreement is adjudicated by a third annotator.
>
> > Q5. It would have been nice if there was an analysis how relevant/aligned performance on the test points was to performance on the downstream capabilities outlined in the paper.
>
> Guided by the capability gaps revealed by MedMT, we implemented targeted capability enhancements. **Including (i) long-text comprehension and memory in extended dialogues, and (ii) detection of, and appropriate responses to, implicit user requests.** Due to safety considerations, we are not able to disclose the results of online A/B experiments at this time.
>
> > Q6. Please report the consistency rate for all the 7 models for which the human evaluation was conducted.
>
> We have supplemented the remaining models. Unfortunately, Claude-4-Sonnet cannot be provided because this model is prohibited. But it should be similar to Claude-4-Opus.
>
> | Consistent rate   | Auto-Eval without ATPs| Auto-Eval with ATPs|
> |-------------------|-----------------------|--------------------|
> | GPT-5             | 40.33%                | 85.40% ±1.2        |
> | Claude-4-Opus     | 39.68%                | 86.97% ±0.8        |
> | OpenAI o3         | 37.98%                | 87.03% ±1.1        |
> | Gemini-2.5-Pro    | 41.09%                | 91.94% ±0.6        |
> | GPT-4o            | 39.91%                | 86.99% ±1.0        |
> | GPT-4.1           | 40.93%                | 88.96% ±0.8        |
>
> > Q7. some relevant recent work on Multi turn conversations is not cited: MMMT-IF.
>
> We have cited this paper in the section 2.2.
>
> > Q8. Does the benchmark task that the LLM is asked to do corresponding to a real LLM use case in health care? Are LLMs already applied in this setting?
>
> As noted, **we have deployed large language models in real-world medical scenarios at scale, serving a substantial user base**. Most category types in the MedMT benchmark were distilled from issues observed in production models.
>
> > Q9. Does it add value on top of having two evaluation benchmarks that already exist?
>
> - Firstly, "Multi-person/disease Interference" and "Medical Security Defense" are more specifically targeted at the medical field;
> - Furthermore, "Information Contradictions" and "Ambiguous Requests" are essential due to the rigorous nature of medical field;
> - Then, "Resistance to Contextual Interference" is directly linked to the complexity and uniqueness of online business;
> - Finally, the medical field's inherently long-memory dialogue requires a greater capacity for understanding context.
>
> > Q10. Please clarify how the consistency rate is computed, a simple equation would make it more clear.
>
> $$
>  \text{Consistency Rate} = \frac{C}{T}
> $$
> where $C$ is the number of instances matched, and $T$ is the total number of samples.
>
> > Q11. Have you considered the bias that may come from evaluating Gemini 2.5 Pro with Gemini 2.5 Pro also as the judge.
>
> We presented a detailed comparison of the agreement rate between Gemini-2.5-Pro ​​and human assessments when evaluating responses from different models. It shows that Genmini-2.5-Pro ​​did not exhibit self-bias with the the fine-grained test points.
>
> |       Models      | Consistent rate |
> |-------------------|-----------------|
> | GPT-5             | 89.50%          |
> | Claude-4-Opus     | 91.25%          |
> | OpenAI o3         | 92.25%          |
> | Gemini-2.5-Pro    | 93.00%          |
> | GPT-4o            | 94.50%          |
> | GPT-4.1           | 91.25%          |

---

> ### Author Response · Authors · 2025-11-27
> **Kindly Requesting Your Thoughts on Our Response**
>
> Dear Reviewer,
>
> We sincerely appreciate your thorough feedback on our paper. As the rebuttal period is coming to a close, could you please confirm if our response has adequately addressed your concerns? If you have any remaining suggestions or would like to continue the discussion, we would be more than happy to make further adjustments.

---

### Official Review · Reviewer_v846 · 2025-10-31

**Soundness:** 3
**Presentation:** 2
**Contribution:** 3
**Rating:** 4
**Confidence:** 4

**Summary:**

Authors introduce MedMT-Bench to evaluate the long multi-turn instruction-following capabilities of LLMs on medical domain in five different dimensions as: (i) long-context memory and understanding, (ii) resistance to contextual interference, (iii) self-correction, affirmation and safety defense, (iv) instruction clarification; and (v) multi-instruction response with interference. Their evaluation data is generated by using a multi-agent pipeline and refined by medical experts for realism, resulting with 400 manually curated test samples averaging 22 turns.  Authors use an LLM-as-a-Judge (LaaJ) metric and evaluate 17 different LLM with both closed- and open-source version, where all score below 60%; revealing persistent weaknesses in long-context understanding and safety compliance.

**Strengths:**

- Authors focus on multi-turn evaluation in the medical domain, which is both timely and critically important.
- Authors conduct comprehensive analysis across different LLMs and share insightful observations, providing possible opportunities for future work in both pure LLM research and medical domain.
- Proposed data generation pipeline combines multi-agent synthesis with manual expert editing, resulting in 400 high-quality samples; which is promising for ensuring the reliability and accuracy of the benchmark.

**Weaknesses:**

- The main limitation of the paper is its heavy dependence on LaaJ during evaluation, which poses additional concerns for specialized domains like medicine. Although the authors conduct human analysis, there is no guarantee that the LaaJ model (Gemini in this case) has sufficient domain knowledge to provide accurate judgments, especially when tasks increase in difficulty. In such situations, LaaJ may prove untrustworthy, especially for medical factuality. A multi-agent LaaJ approach could serve as an interim solution, where knowledge gaps in one model are compensated by others. Furthermore, authors do not provide robustness analysis for Gemini's evaluation performance, such as multiple evaluation runs with reported standard deviations, or cross-validation using different LLM judges with score correlation analysis. Additionally, since LLMs typically find ranking easier than absolute scoring, a ranking-based LaaJ approach might prove more reliable.

- Another concern is evaluation cost and practicality. Given that Gemini is a proprietary model requiring paid API usage, and the benchmark involves long-context, multi-turn interactions, the token consumption per evaluation can be substantial. The paper would benefit from a cost analysis or discussion of the feasibility of large-scale use.

- For practical deployment, such an environment should be user-friendly and easy to debug (for example for understanding LLM failure cases). How do the authors ensure that reviewers can confidently assess the evaluation environment's ease of use?

**Questions:**

1. Did the authors conduct statistical significance testing? Are the results statistically significant?
2. How sensitive are the results to different prompt structures, temperature variations, or LaaJ model selection?
3. How were safety-related turns (e.g., mental health, self-harm prompts) annotated and verified?
4. How do the authors ensure medical factual accuracy, given that LLMs are prone to hallucination?

---

> ### Author Response · Authors · 2025-11-21
> **Thank you so much for your comments! We have answered all your questions point by point.**
>
> Thank you so much for your comments! Thanks to your helpful suggestions, we have supplemented and improved our paper. Below are our responses to the questions you raised.
>
> > Q1. authors do not provide robustness analysis for Gemini's evaluation performance
>
> We conducted three replicate experiments; the results and their variance are as follows. With fine-grained TPs, the standard deviation is small (±0.1 to ±0.6), indicating stable outcomes across evaluation runs.
>
> | Model Names         | Round1 | Round2 | Round3 | Avg           |
> |---------------------|--------|--------|--------|---------------|
> | GPT-4o (2024-11)    | 34.75  | 35.25  | 34.50  | 34.83 ±0.3    |
> | GPT-4.1 (2025-04)   | 41.75  | 42.00  | 42.25  | 42.00 ±0.2    |
> | Gemini-2.5-Pro      | 50.50  | 50.75  | 49.50  | 50.25 ±0.5    |
> | OpenAI o3 (2025-04) | 48.75  | 49.00  | 48.50  | 48.75 ±0.2    |
> | Claude4-Sonnet      | 52.25  | 52.75  | 53.50  | 52.83 ±0.5    |
> | Claude4-Opus        | 54.00  | 54.00  | 53.75  | 53.92 ±0.1    |
> | GPT-5-high          | 60.50  | 59.25  | 60.50  | 60.08 ±0.6    |
>
> In addition, we presented a detailed comparison of the agreement rate between Gemini-2.5-Pro ​​and human assessments when evaluating responses from different models. The results show that, Genmini-2.5-Pro ​​did not exhibit self-bias in the correctness of its own responses.
>
> |       Models      | Consistent rate |
> |-------------------|-----------------|
> | GPT-5             | 89.50%          |
> | Claude-4-Opus     | 91.25%          |
> | OpenAI o3         | 92.25%          |
> | Gemini-2.5-Pro    | 93.00%          |
> | GPT-4o            | 94.50%          |
> | GPT-4.1           | 91.25%          |
>
> > Q2. Another concern is evaluation cost and practicality.
>
> We have added a cost analysis for a single evaluation to the **Appendix D.1**. On average, the evaluator processes 424 input tokens and produces 127 output tokens per evaluation. The estimated cost per evaluation is approximately USD 0.72.
>
> > Q3. How do the authors ensure that reviewers can confidently assess the evaluation environment's ease of use?
>
> We have uploaded a minimal evaluation script as a supplementary file. To reproduce results, users only need to supply the API key of judge model; the script will then generate the evaluation outcomes for the specified model and output all required summary statistics.
>
> > Q4. How sensitive are the results to different prompt structures, temperature variations, or LaaJ model selection?
>
> We tested the sampling temperature and to prompt-structure variations. See **Appendix C.5** for detailed. In terms of the results, even changing the temperature or prompt structures did not cause significant fluctuations.
>
> | Model Names         | tp-1.0 | tp-0.5 | tp-0  | Avg (tp)     | st-v1 | st-v2 | st-v3 | Avg (st)     |
> |---------------------|--------|--------|-------|--------------|-------|-------|-------|--------------|
> | GPT-4o (2024-11)    | 34.50  | 35.25  | 34.75 | 34.83 ±0.3   | 34.75 | 35.50 | 35.50 | 35.25 ±0.4   |
> | GPT-4.1 (2025-04)   | 41.75  | 42.25  | 41.75 | 41.92 ±0.2   | 41.75 | 42.25 | 42.25 | 42.08 ±0.2   |
> | Gemini-2.5-Pro      | 51.50  | 49.50  | 50.50 | 50.50 ±0.8   | 50.50 | 49.00 | 50.75 | 50.08 ±0.8   |
> | OpenAI o3 (2025-04) | 48.25  | 49.50  | 48.75 | 48.83 ±0.5   | 48.75 | 47.75 | 49.50 | 48.67 ±0.7   |
> | Claude4-Sonnet      | 52.00  | 52.75  | 52.25 | 52.33 ±0.3   | 52.25 | 52.25 | 52.75 | 52.42 ±0.4   |
> | Claude4-Opus        | 54.50  | 53.25  | 54.00 | 53.92 ±0.5   | 54.00 | 53.50 | 54.75 | 54.08 ±0.5   |
> | GPT-5-high          | 59.75  | 59.75  | 60.50 | 60.00 ±0.4   | 60.50 | 59.50 | 60.00 | 60.00 ±0.4   |
>
> > Q5. How were safety-related turns annotated and verified?
>
> Our handling of safety issues in synthesized multi-turn contexts falls into two scenarios:
>
> Scenario 1 (incidental unsafe content during synthesis): This data will be discarded.
>
> Scenario 2 (targeted evaluation of safety defense): In this section, we will deliberately construct warning messages that violate system-level prohibitions to examine the model's defense capabilities.
>
> > Q6. How do the authors ensure medical factual accuracy, given that LLMs are prone to hallucination?
>
> MedMT is designed primarily to assess **instruction-following**. We created this benchmark because, in real-world deployments, we observed persistent issues in these settings. For factual accuracy, existing benchmarks such as HealthBench can provide effective evaluation. We are concurrently developing this factuality-focused component and plan to release it in the near future.

---

> ### Author Response · Authors · 2025-11-27
> **Kindly Requesting Your Thoughts on Our Response**
>
> Dear Reviewer,
>
> Thank you again for your insightful comments on our paper. With the rebuttal deadline approaching, we would be grateful if you could let us know whether our response has addressed your main points. Should you have any additional feedback or require further revisions, we are eager to engage in any necessary improvements.

---

### Official Review · Reviewer_uAto · 2025-11-04

**Soundness:** 2
**Presentation:** 3
**Contribution:** 3
**Rating:** 2
**Confidence:** 3

**Summary:**

The papeer proposes MedMT-Bench, a benchmark for evaluating large language models in clinical multi-turn dialogue. It focuses on realistic medical consultation and treatment scenarios. The authors introduce multiple evaluation dimensions addressing long clinical conversation challenges. Data are generated using a multi-agent synthetic pipeline followed by expert medical editing. Evaluation uses fine-grained atomic test points, achieving up to 92.18% agreement with human judgments.

**Strengths:**

1. Medical multi-turn conversations are both high-risk and frequent in real clinical workflows. The paper is well-motivated by identifying that existing medical benchmarks primarily test static knowledge via single-turn or short-dialogue formats.
2. The benchmark is structured around the entire diagnosis and treatment process from pre-diagnosis, to post-diagnosis. This improves the validity and realism of the scenarios.
3. The paper introduces an atomic test points mechanism for its LLM-as-judge protocol that does enhances evaluation reliability.

**Weaknesses:**

1. While the benchmark covers 24 medical departments, the main results in Table 3, 4 only report aggregate performance. There is no analysis of which departments are more or less difficult, which should be common for medical benchmark studies and provide more clinical insights.
2. Lack of modality ablation study: the paper presents results for text, image, and merged subsets but does not provide a quantitative breakdown of which capabilities are most impacted by the image modality. The analysis remains at a high level for model performance.
3. Benchmark sample size 400 may be small relative to task diversity.
4. LLM-based evaluation may reinforce model biases. Also, safety critical metrics false positives not measured in the experiments.
5. Some texts in image are too small to recognize (Figure 5).

**Questions:**

1. The multi-agent synthetic data generation may inherit biases (e.g., style, over-sensitivity, or error propagation) from the models used for synthesis. Can you describe how to address this risk?
2. The atomic test points are central to the evaluation , but the main text provides only one brief example ("check whether... Asiplin"). Could you clarify the full methodology to faciltate reader understanding?

---

> ### Author Response · Authors · 2025-11-21
> **Thank you so much for your comments! We have answered all your questions point by point.**
>
> Thank you so much for your comments! Thanks to your helpful suggestions, we have supplemented and improved our paper. Below are our responses to the questions you raised.
>
> > Q1. There is no analysis of which departments are more or less difficult.
>
> We further explored the statistical indicators of different models in different departments in **Appendix C.4**. The results showed no significant trend across different departments. It is worth noting that almost all models performed worse in nephrology than in other departments, which may indicate the shortcomings of existing models in this department.
>
> > Q2. Lack of modality ablation study.
>
> We update the **Figure 7** that each bar consists of two parts: the solid part represents the performance of the text subset, and the transparent part represents the performance of the whole set after combining the image modal subset. After incorporating the image modality, almost all models have seen improvement in several dimensions, including anaphora resolution, information contradiction, and multi-person interference. This is mainly because these problems are concentrated in the image information under the image modality, which narrows the scope of the model's investigation.
>
> > Q3. Benchmark sample size 400 may be small relative to task diversity.
>
> Our benchmark was designed to identify and evaluate persistent instruction-following challenges observed in existing models across diverse medical scenarios. **It is intentionally a challenge set: each example is carefully curated, and even with multi-agent automated synthesis followed by verification, producing a single usable item typically requires reviewing approximately 20 synthesized candidates.** Specifically, our primary focus is on five categories of instruction-following issues that arise in long, multi-turn dialogues. Additional dimensions such as department and data modality were included to introduce controlled variability and to probe whether these factors exert any supplementary effects.
>
> > Q4. LLM-based evaluation may reinforce model biases. Also, safety critical metrics false positives not measured in the experiments.
>
> To verify the robustness of the LLM-based assessment, we conducted detailed experiments in **Appendix C.5**. After multiple runs, the volatility of the LLM evaluation was limited to within ±0.6. Furthermore, increase temperature or changing the structure of the evaluation prompt, the overall volatility was limited to within ±0.8.
>
> In addition, we presented a detailed comparison of the agreement rate between Gemini-2.5-Pro ​​and human assessments when evaluating responses from different models. The results show that, Genmini-2.5-Pro ​​did not exhibit self-bias in the correctness of its own responses.
>
> |       Models      | Consistent rate |
> |-------------------|-----------------|
> | GPT-5             | 89.50%          |
> | Claude-4-Opus     | 91.25%          |
> | OpenAI o3         | 92.25%          |
> | Gemini-2.5-Pro    | 93.00%          |
> | GPT-4o            | 94.50%          |
> | GPT-4.1           | 91.25%          |
>
> Finally, we also analyzed two recurring patterns in the mis-judgments made by LLM:
> - Over‑generalization: The judge extrapolates beyond the stated rubric, leading to an over‑call of non‑compliance.
> - Instruction comprehension challenges: The judge misreads nuanced constraints, incorrectly concluding a violation when none exists.
>
> As base model capabilities improve, these error modes are expected to decrease further.
>
> > Q5. Some texts in image are too small to recognize (Figure 5).
>
> We have optimized **Figure 6** by enlarging the text and reducing the spacing between sub-figures.
>
> > Q6. The multi-agent synthetic data generation may inherit biases. Can you describe how to address this risk?
>
> Our core objective is to evaluate instruction-following in medical settings, verifying whether current models possess the foundational capabilities required for application in these scenarios. To ensure data diversity, we used a range of models, such as **GPT-5, Gemini-2.5-Pro, and Claude**, when generating the synthetic data. To ensure clinical plausibility and relevance, every item in the benchmark underwent **professional medical review**, focusing on whether the qeustion is reasonable.
>
> > Q7. The atomic test points are central to the evaluation. Could you clarify the full methodology to faciltate reader understanding?
>
> We have expanded **Section 3.3** to describe in greater detail the generation and validation pipeline for test points, and we have added two more detailed illustrative examples. Concretely, given a synthesized, task-specific multi-turn context, we generate in the final turn an instruction-following question targeting a specific evaluation dimension and, concurrently, its corresponding test point. Each multi-turn sample is paired with a corresponding test point.

---

> ### Author Response · Authors · 2025-11-27
> **Kindly Requesting Your Thoughts on Our Response**
>
> Dear Reviewer,
>
> We truly appreciate the time and effort you've put into reviewing our paper. As the rebuttal period is nearing its close, we wanted to kindly ask if our response has addressed your key concerns. If you have any additional suggestions or would like further clarification, we are more than happy to continue the discussion and make any necessary revisions.

---

### Note · Authors · 2025-12-26

I have read and agree with the venue's withdrawal policy on behalf of myself and my co-authors.